# Student Evaluation of Distance Learning during the COVID-19 Pandemic: A Cross-Sectional Survey on Medical, Dental, and Healthcare Students at Sapienza University of Rome

**DOI:** 10.3390/ijerph191610351

**Published:** 2022-08-19

**Authors:** Marco Lollobrigida, Livia Ottolenghi, Denise Corridore, Gianluca Pingitore, Cecilia Damiano, Giorgio Serafini, Alberto De Biase

**Affiliations:** 1Department of Oral and Maxillofacial Sciences, Sapienza University of Rome, 00161 Rome, Italy; 2Department of Cardiovascular, Endocrine-Metabolic Diseases and Aging, Istituto Superiore di Sanità, 00161 Rome, Italy

**Keywords:** COVID-19, medical education, dental education, distance learning, online learning, university learning

## Abstract

The COVID-19 pandemic has had a deep impact on university education, necessitating an abrupt shift from face-to-face learning to distance learning (DL). This has created new challenges, especially for those courses in which practical activities and internships are integral parts of the education program. The aim of this study was to assess the impact of DL on the study progress of a population of pregraduate students of medicine, dentistry, and healthcare professions. The survey was administered through an anonymous questionnaire by sharing a Google Forms link. Demographic data and educational background information were collected to obtain a profile of the participants. Different aspects of DL were investigated, including availability of digital devices, quality of connection, and environmental conditions; other questions focused on the effects of DL on students’ progress and professional maturation. Measures of association were also calculated using the chi-squared test, Cramer V, and Somers D. Among the 372 who participated, the results showed that students had a positive attitude toward online classroom and that DL did not substantially affect their progress. Most of the associations were statistically significant, also highlighting the effect of the degree course on the responses. Some critical issues clearly emerged, however, including the lack of adequate devices and environmental conditions due to economic disparity, poor relationships, suspension of internship programs, and clinical training. The results suggest that DL cannot be considered as a substitute for classroom-based medical education outside an emergency context.

## 1. Introduction

The COVID-19 pandemic has enormously impacted the lives of billions of people around the world, necessitating severe containment measures to limit the spread of infection. These included the suspension or drastic reduction of social activities such as work, education, productive processes, culture, recreation, and sport. In Italy, the first isolated cases of COVID-19 infection were recorded at the end of January 2020, though an earlier spread was hypothesized in the prepandemic period [1]. Later outbreaks were highlighted in northern towns, spreading soon to the next regions and then to the entire nation. Consequently, on 9 March 2020, a state of total lockdown was decreed [2].

The pandemic had important repercussions on university education, where face-to-face lessons and other didactic activities were suspended for indefinite periods. Tertiary institutions had to rapidly adapt to the changes caused by the pandemic, and, after an initial period of uncertainty, some teachers started to share didactic material, others opted for asynchronous teaching, and others for synchronous virtual classrooms on dedicated platforms (Google Meet, Zoom, etc.) [3,4]. Following this phase of heterogeneous approaches, distance learning (DL) with scheduled live-streamed lectures became the standard teaching modality of universities. DL assured direct communication between teachers and students, but also created many challenges [5,6], both logistical and pedagogical, in particular for health science courses where internships and practical training are integral parts of the education programs [7,8]. Moreover, even with the recovery of face-to-face learning, limited access to the clinics and the interpersonal distance significantly impacted student training and clinical skills acquisition.

Considering the implications of the COVID-19 pandemic on medical education, the objective of this study was to collect information on the impact of DL on the careers and maturation of a sample of pregraduate students from different university programs, including the medicine, dentistry, and healthcare professions.

## 2. Materials and Methods

A cross-sectional study was conducted at Sapienza University of Rome among students of medicine and surgery, dentistry, and health professions through an online survey. Participation was anonymous and voluntary, and the survey description specified that all data collected would be used for research purposes only. The questionnaire was prepared on the Google Forms (Google LLC, Mountain View, CA, USA) platform and shared via academic communication channels (mailing lists) from May to June 2021.

### 2.1. Survey Design

The survey was first designed following a literature search on the effects of COVID-19 on medical education. A preliminary draft was then reviewed by a focus group that included student and teacher representatives from the university board. The final draft was structured in two parts, for a total of 32 elements. In the first section, demographic data were collected including age, sex, faculty, degree course, and previous DL experiences. In the second section (Figure 1), different aspects of DL were investigated: some questions regarded availability of digital devices, quality of internet connection, and adequateness of home environmental conditions. Other questions focused on the impact of online education on study progress and professional maturation. The remaining elements of the questionnaire investigated students’ opinion on the advantages, disadvantages, and future opportunities of DL. The survey consisted of multiple-choice questions, including single-answer Likert-type questions (ranging from strongly disagree to strongly agree), and open-ended questions. Some questions were conditional.

### 2.2. Participants

All students of medicine, dentistry and healthcare professions (1 to 3 years for healthcare professions degrees and 1 to 6 for medicine and dentistry degrees) were eligible to participate with the exclusion of out-of-course students. According to the official data provided by the University, 4128 students were eligible at the time of the survey.

### 2.3. Statistical Analysis

The answers received were assessed through the Google Forms percentage report. For the present study, descriptive analysis was carried out, and several measures of association were performed, including the chi-square test, Cramer’s V, and Somers’s D [9,10,11,12]. The chi-square test allows for the identification of the association between two categorical variables. The null hypothesis is that there is no relationship. Cramer’s V measures the strength of the association between the two variables, ranging from 0 to 1, where 1 indicates a strong association. Such tests analyzed the associations between the course study and the answers given by students to questions 8, 9, 12, and 14. Somers’ Delta (Somers’ D) is a measure of the association between pairs of ordinal variables. More specifically, asymmetric Somers’ D measures how much the prediction of the dependent variable improves, based on knowing a value of the independent variable. It ranges between −1 (all pairs disagree) and +1 (all pairs agree); values tending toward −1 or +1 suggest that the model has good predictive ability. This test was used to study the association between questions 6, 11, 13–16, 19, and 20. In all the analyses, a *p* value of <0.05 was considered as statistically significant. Statistical analyses were performed using the software Stata version 16.0 (Stata Corporation, College Station, TX, USA).

## 3. Results

A total of 372 responses were collected, with a response rate of 9.16%. Characteristics of the respondents are shown in Table 1. The students who participated had a mean age of 23 (±4) years and 258 (69%) were women. Among students, 47% were enrolled in dentistry (first-year students, 24%), 37% in medicine and surgery (first-year students, 10%), and 16% in healthcare professions (first-year students 72%). By comparing the demographic characteristics of the respondents with the official data provided by the university (https://statistiche.uniroma1.it/portale/extensions/Portale_Pubblico/Portale_Pubblico.html, accessed on 7 February 2022), we concluded that the respondents were representative of the student population for age and sex, while the relative number of students from each degree course was less represented, with a predominance of students from dentistry.

In the whole sample, most students were off-site (59%). Moreover, 12% of students declared previous experience with DL before the COVID-19 pandemic.

The complete list of questions and response frequencies can be found in the Appendix A. Most students considered themselves sufficiently prepared for using the informatic platforms at the beginning of the pandemic (Q1). Twenty percent of students did not have adequate digital devices for DL, and some could not afford the purchase of new devices (Q2). To follow online lessons, 86.6% of the participants used a PC, 9.1% a tablet, and 4.3% used smartphones (Q3). With respect to the quality of internet connection, only 21.8% reported having an excellent quality connection, 52.9% a good connection and 17.7%, and 7.5% a fair or poor connection (Q4). Overall, 31.5% of students reported having experienced difficulties in following the lessons due to connection issues (Q5).

As for environmental conditions and their impact on learning, more than 2 out of 10 students did not have an adequate environment (Q6), while nearly half reported difficulties or distractions due to the presence of cohabitants, roommates, or family members (Q7).

Overall, DL did not have a significant effect on students’ performance in terms of exam results (Q8) or study progress (Q9), as confirmed by the university records (data not presented). However, a negative impact (answers “definitely yes” and “more yes than no”) on professional development (Q10) and motivation (Q11) was reported by 48.4% and 55.3% of the students, respectively. Similarly, 77.4% felt that the suspension of internship and practical activities produced significant gaps in their preparation (Q12).

In the students’ opinion, teachers had a good attitude toward online teaching (Q13) and telematic exams were suitable for a correct evaluation of their preparation (Q14). However, according to 59.4% of respondents, DL reduced the participation and interaction with teachers and course mates (Q15) while, on the other hand, it favored the sharing of materials (Q16).

With respect to the risks related to spending several hours in front of electronic devices, 59.41% of the participants reported having spent more than 6 h per day in front of the screen due to DL (Q17) and having experienced disorders, mainly related to visual fatigue and tiredness (Q18, see also Figure 2). Despite this, 46% of the students considered DL less stressful than face-to-face lessons (Q19).

Overall, 63.4% of the sample positively judged DL (Q20).

The report of the answers to the last multiple-choice questions (Q21–Q23) is shown in Figure 2. There was broad consensus among the respondents that DL reduced the need for travel, saved time and money, and allowed for greater flexibility (Q21). Among the disadvantages, low interaction with teachers and colleagues, home distractions, and the quality of internet connection were highlighted (Q22). In the students’ opinions, seminars, frontal lessons, elective didactic activities, and group works are activities that could still be conducted in a telematic mode in a postpandemic scenario (Q23). However, it is also important to note that 19.1% of students believed that the DL modality would not be suitable for any activity (see Appendix A).

When analyzing the measures of association considering the degree course (Table 2), most of the associations were found to be statistically significant, highlighting the effect that the degree course had on students’ responses.

As for the association between variables, all the tested associations were found to be statistically significant (Table 3).

## 4. Discussion

During the lockdown periods and the reopening phase for quarantined classes, synchronous online courses have guaranteed the continuity of educational programs, but at the same time revealed critical issues with specific nuances depending on the level and area of education [13]. Some concerns specifically regarded medical education, where the consequences of poor and disengaged learning may be significant [14].

As already observed in another survey [15], the results of this study showed that students were digitally prepared for the shift from classroom to online lessons, though DL was uncommon before the pandemic for most of them. In recent years, the diffusion of e-learning in Italy has been considerably slower and more difficult than in other countries due to the deep-rooted cultural tradition of classroom training. A recent report, published in 2021 by the digital learning platform “Preply”, assessed the e-learning-index of thirty different countries. In this study, Italy ranked 27th, confirming the difficulty in promoting digital education, also due to the still-limited national broadband and fiber coverage (https://preply.com/it/d/e-learning-index/, accessed on 2 March 2022). However, more than the digital attitude, the availability of digital devices (Q2 and Q3) as well as the quality of internet connection (Q4 and Q5) represented barriers for creating an adequate DL environment. The pandemic further accentuated social inequalities, worsening the economic conditions of many families [16,17], with the result that some students could not afford adequate digital devices [18]. Following the allocation of numerous public funds by the state, regions, and municipalities, all Italian schools have taken action to make laptops, notebooks, and tablets available to school students on a free loan basis. In the context of a digital solidarity initiative (https://solidarietadigitale.agid.gov.it/iniziative/, accessed on 2 March 2022), schools also suggested students’ families contact their telephone operator to be updated on the opportunities made available for the free increase in internet traffic. Conversely, universities’ initiatives were jeopardized, and a percentage of students had to follow online lessons on their smartphones (Q3).

Further concerns are related to the domestic environment of students. More than 20% of respondents reported to live in environments inadequate for DL (Q6), while almost half reported having experienced difficulties or distractions related to the presence of family members and/or roommates (Q7). It has been demonstrated that the physical environment can significantly influence cognition, emotions, and behavior [19], and that distractions and small interruptions can have detrimental effects on the learning process [20]. In this respect, low-income students have suffered most from the impact of the pandemic due to precarious home conditions and lack of adequate spaces and tranquility [21].

In accordance with previous reports [22,23,24], students’ overall opinion of DL was positive in the context of the COVID-19 pandemic, and teachers’ attitudes toward online lessons were positively viewed (Q13). DL has not had a negative impact in terms of exam marks and study progress for most respondents (Q8 and Q9), as also indicated by the official data provided by the university and thus confirming the reliability of students’ responses; notwithstanding, students reported having suffered a learning loss (Q10). Students then distinguished between a mere performance in terms of marks and passing exams, and a broader concept of learning, which suffered from the limits and criticalities imposed by DL. Although DL can be advantageous in certain aspects, students’ active participation seems to have been affected (Figure 2). Moreover, the suspension of internship programs and practical activities had important repercussions because practical learning activities, laboratories, and hospital internships cannot be transferred in a DL context, and dental training on manikins alone is considered insufficient [25].

When testing for the presence of associations, considering the course degree, some interesting results were found. Changes in performance (expressed as average of grades), impact of DL on study progress, and impact of the suspension of internships on the professional preparation were significantly associated with the course of study, highlighting that the degree course influenced students’ responses. In particular, for the students of healthcare professions, DL had less of an effect on their grade point average and their study progress, compared with students enrolled in dentistry and medicine and surgery. On the other hand, for this same group of students, the suspension of internships produced the most significant gaps in professional preparation. This could be related to the shorter duration of healthcare profession courses (three years) compared with those for medicine and dentistry (six years), denoting a more severe perceived impact of internship suspension.

The abrupt introduction of DL implied poor attention regarding the possible risks of spending several hours in front of electronic devices and, in general, students’ mental health [26]. It is interesting to note that while precise instructions exist about video terminal workers, to date, no guidelines have been developed in this respect for university students. Regarding primary and secondary schools, the Ministry of Education affirmed that it is necessary to avoid making students spend too much time online, seeking a balance between on-screen and traditional off-screen activities. In this survey, most respondents reported to have spent more than 6 h per day in front of their digital devices (Q17) and to have experienced physical health issues due to DL (Q18), adding this to the other mental health issues related to the pandemic and isolation in general [27], and environmental stressors [28]. This advocates for guidelines addressed to the different levels of education and with specific recommendations with respect to the maximum time to spend in front of digital devices.

According to the students’ opinions, the main advantages of DL are represented by the possibility to guarantee teaching continuity in emergency situations, time flexibility, and the easy sharing of educational material. From this perspective, DL can be an opportunity to ensure access and study continuity in special cases, such as students’ long absences. On the other hand, limits of DL have clearly emerged, primarily related to the need for up-to-date computer equipment and adequate domestic environmental conditions and the lesser interaction between students and teachers and among students [29,30]. More importantly, DL could not replace practical activities such as internships and hands-on training. These results are in accordance with those of similar cross-sectional studies from other countries. A multicenter cross-sectional study among German medical students [31] revealed similar concerns about poor exchange between fellow students and with teachers as well as the lack of practical training. Additionally, a proportion of students was unable to participate in DL due to the lack of adequate devices. As a result, the preferred form of education was on-site teaching.

The analyses also allowed for the identification of the association between some of the aspects covered in the questionnaire. Students’ motivation and participation in the lessons were influenced by the teachers’ attitude toward DL and by the sharing of teaching material. Moreover, students’ overall opinion on the DL experience was strongly influenced by the perceived stress level (compared with face-to-face teaching), the home environment, and students’ experience with oral exams in telematic mode.

The results of this study suggest that, despite its technical feasibility, DL is not an adequate substitute for frontal lessons and clinical training, which represent the cornerstone of medical education. The relative benefits reported by some students do not justify the limitations that have emerged. Future research should, however, explore the potential of DL as a complementary tool outside of an emergency. Long-lasting observational studies should evaluate the differential impact of entirely face-to-face courses and hybrid (face-to-face and DL) courses on students’ preparation and satisfaction.

## 5. Conclusions

According to the students’ opinions, despite some benefits, DL presents important drawbacks that limit its use in medical education in the long term. Fundamental aspects including equality, human relationships, and adequate clinical training cannot be guaranteed by DL and this can demotivate students. In the postpandemic period, however, in-depth thinking about which activities can take advantage of digital tools would be beneficial, avoiding merely transferring traditional didactic approaches to a digital context. Universities should then develop guidelines to make a conscious use of DL, where this proves to be useful, and guarantee essential digital devices to students who cannot afford them. While waiting for new didactic approaches to be evaluated, traditional medical education remains a cornerstone approach.

## Figures and Tables

**Figure 1 ijerph-19-10351-f001:**
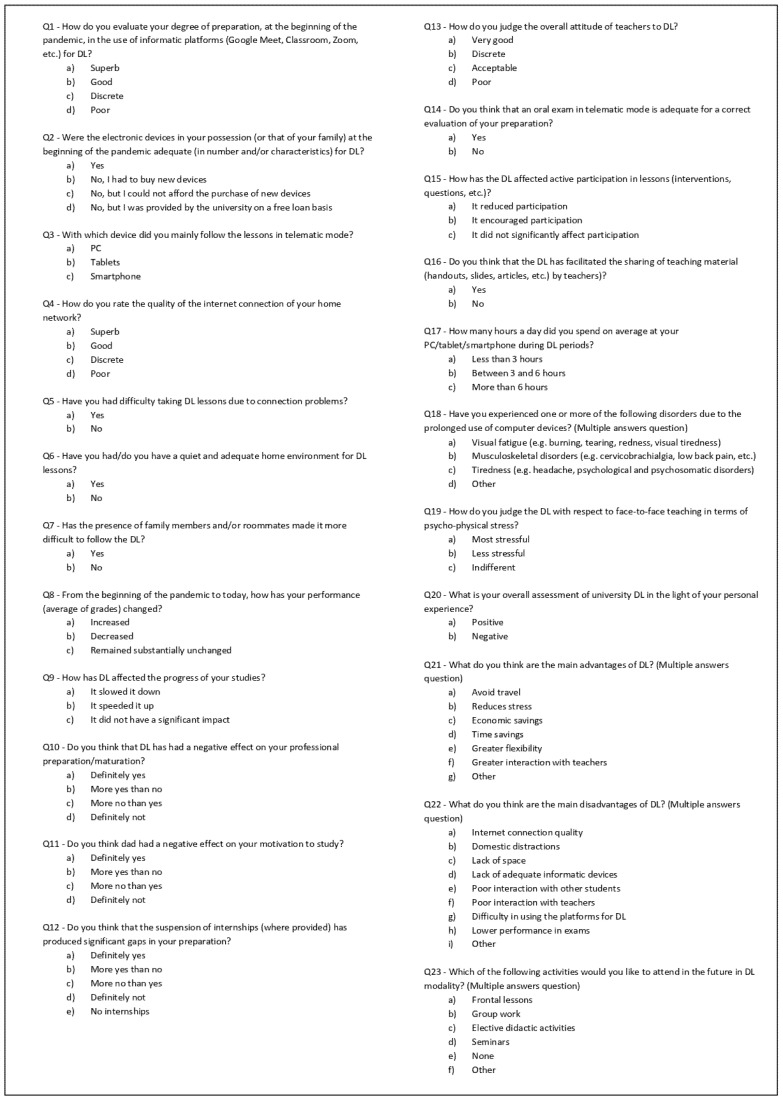
Questionnaire on DL.

**Figure 2 ijerph-19-10351-f002:**
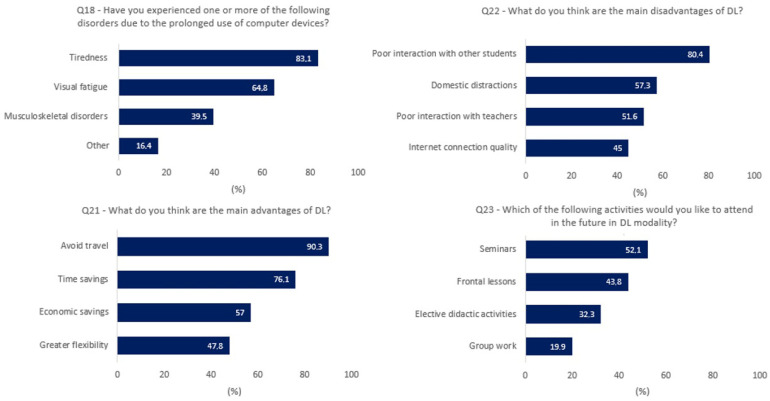
Report of the answers for multiple-choice questions Q18 and Q21–Q23 (most frequent answers only; for complete data see Appendix A).

**Table 1 ijerph-19-10351-t001:** Characteristics of the respondents in the whole sample and by degree course. Values are presented as absolute number and column percentage (%).

Parameter	All, N = 372	Medicine and Surgery N = 139	Dentistry N = 173	Health Professions N = 60
Age (mean, SD)	23.02 (3.7)	22.7 (2.7)	23 (3.3)	23.7 (6.1)
Female	258 (69.3)	94 (67.6)	119 (68.8)	45 (75)
Male	113 (30.4)	45 (32.4)	53 (30.6)	15 (25)
Other	1 (0.3)	0	1 (0.6)	0
Year of enrollment				
First-year	98 (26.3)	14 (10.1)	41 (23.7)	43 (71.7)
Second year onward	274 (73.7)	125 (89.9)	132 (76.3)	17 (28.3)
Off-site student	221 (59.4)	77 (55.4)	107 (61.9)	37 (61.7)
Working student	69 (18.6)	16 (11.5)	32 (18.5)	21 (35)
Experience with DL before COVID-19 pandemic	44 (11.8)	19 (13.7)	18 (10.4)	7 (11.7)
If yes, eventual previous experience with DL (multiple-answer question)				
University	66 (31.9)	31 (27.4)	32 (37.2)	3 (37.5)
Secondary education	13 (6.3)	7 (6.2)	4 (4.7)	2 (25)
Other types of school	128 (61.8)	75 (66.4)	50 (58.1)	3 (37.5)

DL = distance learning.

**Table 2 ijerph-19-10351-t002:** Measures of association considering degree course. Statistically significant associations are in bold.

	Medicine and Surgery	Dentistry	Health Professions	Chi-Squared Test (*p*-Value)	Cramer V
Change in performance since the beginning of the pandemic (average of grades)					
Decreased	19%	22%	2%	0.001	0.16
Substantially unchanged	62%	58%	88%		
Increased	19%	20%	10%		
Impact of DL ^a^ on the studies progress					
Slowed it down	32%	20%	25%	0.001	0.16
No significant impact	50%	43%	58%		
Speeded it up	19%	36%	17%		
Significant gaps in the preparation due to the suspension of internships ^b^					
Yes	90%	78%	98%	<0.001	0.23
No	10%	22%	2%		
An oral exam in telematic mode is adequate for a correct evaluation					
Yes	76%	79%	80%	0.81	0.03
No	24%	21%	20%		

^a^ DL = distance learning ^b^ Those who answered “No internships” where excluded.

**Table 3 ijerph-19-10351-t003:** Measures of association between questions.

Associated Variables	Somers’ D	*p*-Value
Motivation to study of the students given the attitude of teachers toward DL	−0.30	<0.001
Level of students’ participation in lessons given the teachers’ sharing of teaching material	0.26	<0.001
Overall opinion on DL experience given the stress level compared with face-to-face teaching	−0.53	<0.001
Overall opinion on DL experience given a home environment suitable for DL	0.34	<0.001
Overall opinion on DL experience considering an oral exam in telematic mode as adequate for correct evaluation	0.55	<0.001

DL = distance learning.

## Data Availability

The data presented in this study are available on request from the corresponding author.

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
