# Peer review of "Student Evaluation of Distance Learning during the COVID-19 Pandemic: A Cross-Sectional Survey on Medical, Dental, and Healthcare Students at Sapienza University of Rome"

_ijerph, 2022, doi:10.3390/ijerph191610351_

Round 1
Reviewer 1 Report
The authors generally well revised the first review commnets.
However, in Table 2 and 3, the p-value, which is the statistical significant level, is generally presented as '<0.001' rather than '<0.005', so please revise it.
In addition, please review and revised the abbreviation of the journal name in references again.
Author Response
replies in attachment

Reviewer 2 Report
The data from the University on Q8-10 is relevant (not to say important) to the paper. It provides an independent measure how reliable the students' replies are. It is advisable to include it in the Discussion.
Rows 158-160 belong to Discussion, not to Results
Author Response
replies in attachment

Reviewer 3 Report
Dear authors, thank you for resubmision of the paper. You have made significant correctiions, however there are still some doubts from my point of view.
You should carefully go throuh text and exclude sentences which are repeated.
Conclusion is now better, however you should highlight impact of DL to motivation (demotivation) and destimulation which was huge and unfixable for the generation of students.
The traditional method of learning should not be underestimated in the conclusion.
After minor revision you should submit paper for final round.
Author Response
replies in attachment

This manuscript is a resubmission of an earlier submission. The following is a list of the peer review reports and author responses from that submission.
Round 1
Reviewer 1 Report
In the context of the COVID-19 pandemic, it is necessary to evaluate the distance education in the educational field. However, it is judged that it is not appropriate as a research paper to be published in this journal.
In addition, the descriptive statistics of the table need to be rearranged for better readability.
This study simply presents only descriptive statistics, so statistical analysis should be advanced to increase the value of the study. In addition, the research results should be presented with increased readability. In other words, the systematicity of the research thesis must be observed.
The originality of the study is appropriate in terms of educational evaluation in the context of a lot of distance education in the COVID-19 situation. However, the research method must clearly describe the population of participants and provide the rationale for calculating the appropriate number of samples.
This study simply presents descriptive statistics, so a comparative analysis on the differences between distance education and face-to-face education seems necessary.
Since this study calculated descriptive statistics for simple survey questions, it is insufficient as a research paper in this journal. It is necessary to analyze the significant effects of the advantages and disadvantages that appear to students in the health care field due to distance education.
It does not lead to conclusions that are consistent with the research results. It should be supplemented with future research directions based on the main research results.
In the context of COVID-19, many references related to non-face-to-face educational evaluation have been published, so it should be explored more and sufficiently discussed.
Tables should be systematically supplemented for legibility. Frequency and percentage are usually written as 'n (%)'.
Reviewer 2 Report
Line 78 7916 were not participants, they could have been but majority choosed not to. This is the population that the sampled participants were grawn from.
The paper consists of the students oppinion, and this has to be cearly stated along the paper. For example, line 211: according to the students oppinion it is a feasable didactic tool.
There is no indication how the sample compares to the population of the students demographicly so we don't know whether it is representative or not.
The main issue is always how relevant and how generalizable are the findings in a paper. If the results are relevant only to those surveyed, it has little interest to readers except to those involved.
In the paper at hand, a self-selected sample of 378 students responded out of cc. 8000. First question: Is the sample representative of the student population? IF we don’t know, the results are relevant to those 378 and have little scientific merit.
The authors should state and demonstrate if the sample is representative, at least by providing the population demography. Some information about the properties of the students of the Med and Dent schools is helpful so we know if they have some special characteristics.
If the findings can be generalized to the whole student population, then there is place for issues like what it adds to the subject and if the references are relevant.
The other important aspect is that the findings are the students’ views. This aspect is important to the objective of the paper but it would add to know how realistic their views are.
Questions 10 to 12 ask about objective aspects that can be checked in the school records.
Reviewer 3 Report
Thank you for sending me this comprehensive text for reading.
The paper describes in detail the advantages and disadvantages of distance learning during the Covid-19 pandemic.
Results are prested nicely and everything is clear.
The conclusion should have a stronger sentence than you put in the text regarding the clinical part of the studies. (row 213-214). It should be highlighted that online learning is not sufficient for medicine/dentistry programme. It should be stressed that the face to face learning (professor-student) is mandatory in order to achieve the appropriate level of clinical skills.
The main remark is the conclusion, which is not the logical and objective outcome of the presented data. There is a controversy between answers within the text and conclusions. According to the presented data, the feasibility of digital learning is very low. (Which should be reconsidered in the conclusion section.)
The topic is relevant in the field, however, there are many papers published so far with similar data.
The authors presented descriptive statistics. There is a lack of statistical comparisons ( tests). It should be added.
Thank you